# 5-Fluorouracil Suppresses Colon Tumor through Activating the p53-Fas Pathway to Sensitize Myeloid-Derived Suppressor Cells to FasL^+^ Cytotoxic T Lymphocyte Cytotoxicity

**DOI:** 10.3390/cancers15051563

**Published:** 2023-03-02

**Authors:** Yingcui Yang, Mingqing Zhang, Yongdan Zhang, Kebin Liu, Chunwan Lu

**Affiliations:** 1School of Life Sciences, Tianjin University, Tianjin 300072, China; 2Department of Colorectal Surgery, Tianjin Union Medical Center, Tianjin 300121, China; 3Department of Biochemistry and Molecular Biology, Medical College of Georgia, Augusta, GA 30912, USA; 4Georgia Cancer Center, Augusta, GA 30912, USA

**Keywords:** 5-FU, MDSCs, Fas, p53, cytotoxic T lymphocytes, colorectal cancer

## Abstract

**Simple Summary:**

5-Fluorouracil (5-FU), in combination with various therapeutic agents, is the main strategy for patients with high risk of stage 2, and advanced stages of, human colorectal cancer. Although 5-FU-based chemotherapy causes myeloid cell suppression, which has long been considered as a major adverse effect in certain cancer patients, recent studies determined that 5-FU selectively kills myeloid-derived suppressor cells (MDSCs), to increase cytotoxic T lymphocyte activation. However, the molecular mechanism underlying 5-FU’s suppression of MDSCs is incompletely understood. We report here that 5-FU activates p53, to upregulate Fas expression in MDSCs, to increase MDSC sensitivity to FasL-induced apoptosis in vitro. 5-FU therapy upregulates Fas expression, to suppress MDSC accumulation, to increase CTL tumor infiltration in tumor-bearing mice. In human colorectal cancer patients, 5-FU-based chemotherapy suppresses MDSCs and increases CTLs level. Our findings determine that the p53-Fas pathway links 5-FU to MDSC suppression.

**Abstract:**

Myelosuppression is a major adverse effect of 5-fluorouracil (5-FU) chemotherapy. However, recent findings indicate that 5-FU selectively suppresses myeloid-derived suppressor cells (MDSCs), to enhance antitumor immunity in tumor-bearing mice. 5-FU-mediated myelosuppression may thus have a beneficial effect for cancer patients. The molecular mechanism underlying 5-FU’s suppression of MDSCs is currently unknown. We aimed at testing the hypothesis that 5-FU suppresses MDSCs through enhancing MDSC sensitivity to Fas-mediated apoptosis. We observed that, although FasL is highly expressed in T cells, Fas is weakly expressed in myeloid cells in human colon carcinoma, indicating that downregulation of Fas is a mechanism underlying myeloid cell survival and accumulation in human colon cancer. 5-FU treatment upregulated expression of both p53 and Fas, and knocking down p53 diminished 5-FU-induced Fas expression in MDSC-like cells, in vitro. 5-FU treatment also increased MDSC-like cell sensitivity to FasL-induced apoptosis in vitro. Furthermore, we determined that 5-FU therapy increased expression of Fas on MDSCs, suppressed MDSC accumulation, and increased CTL tumor infiltration in colon tumor-bearing mice. In human colorectal cancer patients, 5-FU chemotherapy decreased MDSC accumulation and increased CTL level. Our findings determine that 5-FU chemotherapy activates the p53-Fas pathway, to suppress MDSC accumulation, to increase CTL tumor infiltration.

## 1. Introduction

The chemotherapeutic agent 5-fluorouracil (5-FU) has been the single most commonly used drug for patients with advanced colorectal cancer for the past four decades [1,2]. However, resistance to 5-FU often develops in cancer patients [1]. Increased understanding of the mechanism of 5-FU action has led to the development of strategies to overcome 5-FU chemoresistance, and to improve its efficacy, primarily through combination with other therapeutic agents [2]. However, a major adverse effect of 5-FU is myelosuppression [3,4,5], which is a dose-limiting factor of 5-FU chemotherapy [4].

Recent advances in cancer immunology revealed a critical role of myeloid regulatory cells, particularly myeloid-derived suppressor cells (MDSCs), in immune suppression of the antitumor immune response [6]. MDSCs are a heterogeneous population of myeloid cells of various differentiation stages [7], that are induced under various pathological conditions, including cancer [7]. In human cancer patients and tumor-bearing mice, massive accumulation of MDSCs is a hallmark of cancer progression [8]. One key function of MDSCs is to inhibit activation, and therefore the effector functions, of cytotoxic T lymphocytes (CTLs) [8,9]. In addition, MDSCs modulate a favorable tumor microenvironment for angiogenesis, and tumor growth and progression, through non-immunologic mechanisms [10]. MDSCs, therefore, are widely recognized as potent tumor promoters, that are key targets in cancer therapy [6]. The new advances in understanding of the functions of myeloid regulatory cells indicate that the 5-FU therapy-associated adverse effect of myelosuppression might actually contribute to 5-FU’s antitumor activity, via suppressing MDSCs in cancer patients. Indeed, recent studies have shown that 5-FU selectively induces MDSC apoptosis, resulting in a reversal of immune suppression of CTL function, to inhibit tumor development in tumor-bearing mice [11,12]. However, the molecular mechanism underlying 5-FU’s suppression of MDSCs is largely unknown.

Fas is a death receptor, that is essential for the homeostasis of lymphocytes [13]. Fas is highly expressed on the surface of lymphocytes, and loss of Fas expression or function leads to various diseases, including autoimmune lymphoproliferative diseases and cancer [14,15,16,17,18,19,20,21]. In humans, it was recently discovered that MDSC homeostasis is also regulated by Fas [22]. Consistent with these findings, MDSCs use downregulation of Fas to acquire an increased apoptosis resistance phenotype, to avoid elimination [23]. Therefore, in addition to induction by inflammatory cytokines, increased resistance to Fas-mediated apoptosis is another mechanism underlying MDSC accumulation in the tumor-bearing hosts. It is known that 5-FU activates p53, and p53 can directly activate Fas transcription in tumor cells [24,25]. However, it is unknown whether p53 regulates Fas expression in MDSCs, and whether the p53-Fas pathway links 5-FU to MDSC suppression. We aimed to test the hypothesis that 5-FU suppresses MDSC accumulation through activating the intrinsic p53-Fas pathway, to sensitize MDSCs to FasL-induced apoptosis in a tumor-bearing host. We report here that 5-FU activates p53 expression, to upregulate Fas expression, to increase MDSC sensitivity to FasL-induced apoptosis.

## 2. Materials and Methods

### 2.1. Human Blood Specimens

Peripheral blood specimens were collected form untreated and 5-FU-treated human colon cancer patients in Tianjin Union Medical Center. All samples were collected within 7 days after 5-FU chemotherapy. Studies with human specimens were approved by the Institutional Review Board of Tianjin Union Medical Center (Approval # 0224-2021. Approval date: 3 March 2021) and The Institutional Review Board of Tianjin University (Approval # TJUE-2021-016. Approval date: 1 March 2021). Clinical information of de-identified patients is shown in Appendix A.

### 2.2. Mice 

BALB/c mice were purchased from the Jackson Laboratory (Bar Harbor, ME, USA). All studies involving use of mice were covered by a protocol approved by the Institutional Animal Care and Use Committees of Tianjin University (Approval # TJUE-2021-016. Approval date: 1 March 2021) and Augusta University (Approval # 2008-0162. Approval date: 31 March 2020).

### 2.3. Mouse and Human Colon Tumor Cell Lines

Murine colon tumor cell line CT26, and human colon tumor cell lines RKO, Caco2, Colo201, HT29, DLD1, LS411N, LS174T, SW480, and SW620 were purchased from ATCC (Manassas, VA, USA). The HCT116.WT (clone 8) and HCT116 p53 KO (clone 2) cell lines were kindly provided by Dr. Bert Vogelstein at Johns Hopkins Medical Institutions (Baltimore, MD, USA), via Dr. Phillip Buckhaults at the University of South Carolina (Columbia, SC, USA). All cell lines were tested every 2 months for mycoplasma, and all cell lines used in this study were mycoplasma-free at the time of the study.

### 2.4. J774M Cell Line

The MDSC-like J774M cell line was derived from the J774A.1 cell line. The J774A.1 cell line was obtained from ATCC. J774A.1 cells were stained with CD11b- and Gr1-specific antibodies (Biolegend, San Diego, CA, USA) and analyzed by flow cytometry. The CD11b^+^Gr1^+^ cells were sorted and cultured to establish a stable CD11b^+^Gr1^+^ cell line, termed J774M, as previously described [26].

### 2.5. Recombinant FasL Protein

Mega-Fas Ligand (kindly provided by Dr. Peter Buhl Jensen at Oncology Venture A/S, North Port, FL, USA) is a recombinant fusion protein that consists of three human FasL extracellular domains linked to a protein backbone, comprising the dimer-forming collagen domain of human adiponectin [27]. The Mega-Fas ligand protein was produced as a glycoprotein in mammalian cells using a good manufacturing practice compliant process in Topotarget A/S (Kopenhagen, Denmark).

### 2.6. In Vivo Tumor Model

To establish the CT26 tumor-bearing mouse model, tumor cells (2.5 × 10^5^ cells/mouse) were injected subcutaneously into the right flank of BALB/c mice. The tumor-bearing mice were treated with 5-FU (dissolved in PBS, 25 mg/kg body weight) when tumors reached around 30–50 mm^3^ in size, by i.v. injection once every 2 days, for a total of 6 times.

### 2.7. Human Patient Sample Single-Cell RNA Sequencing (scRNA-Seq) Datasets

Human colorectal cancer patient scRNA-Seq datasets were extracted from the Single Cell Portal (Broad Institute, Cambridge, MA, USA) and GEO database (GSE178341). Cells are annotated according to dataset designation [28].

### 2.8. CRISPR-Based Gene Knockout

Generation of p53 KO cell lines were carried out essentially as previously described [29]. Briefly, HEK293FT cells were co-transfected with psPAX2 (Addgene #12260), pCMV-VSV-G (Addgene #8454) [30], and lentiCRISPRv2 (Genscript, Piscataway, NJ, USA) plasmids containing scramble (GAAGACTTAGTCGAATGAT) or *Trp53*-specific (AGTGAAGCCCTCCGAGTGTC) sgRNA-coding sequences using Lipofectamine (Cat# T101-01, Vazyme, Nanjing, China). Cell culture supernatants (virus particles) were collected and used to infect J774M cells. Stable cell lines were established using puromycin (Cat# BS111, Biosharp, Hefei, China) selection.

### 2.9. Western Blotting Analysis

Western blotting analysis was performed as previously described [29]. Briefly, cells were lysed in total lysis buffer, measured for protein concentration by Bradford Assay Kit (Cat# MA0079, Meilunbio, Dalian, China), and cell lysates were separated in 4–20% SDS-polyacrylamide gels (Bio-Rad, Hercules, CA, USA) and blotted to PVDF membranes (Bio-Rad). The antibodies are: anti-mouse p53 (Cat#2524, Cell Signaling, Danvers, MA, USA; and Cat# AF7671, Beyotime, Beijing, China) and mouse β-actin (Cat# A2228, Sigma-Aldrich, St Luis, MO, USA; and Cat# AF003, Beyotime, Beijing, China). All original Western blotting results are shown in Appendix A.

### 2.10. Flow Cytometry Analysis

For cell lines, cells were firstly harvested, centrifuged, and washed in PBS. The cell pellets were resuspended in 100 μL PBS + 1% BSA and stained with fluorescent dye-conjugated antibodies. All antibodies were obtained from Biolegend: anti-human Fas (Cat#305608), anti-mouse/human CD11b (Cat#101206), anti-human HLA-DR (Cat#307609), anti-human CD8 (Cat#344706), anti-mouse Fas (Cat#152607), anti-mouse Gr1 (Cat#108426), anti-mouse CD8 (Cat#100732), and anti-mouse CD45 (Cat#147712). Stained cells were analyzed by flow cytometry. Data files were analyzed using FlowJo.V10 software.

### 2.11. Cell Death Analysis

J774M cells (1 × 10^5^/well) were cultured in the presence of recombinant FasL protein alone, or in combination with 5-FU (10 μg/mL), for 24 h. Both attached and nonattached cells were harvested, washed in PBS. Propidium iodide (PI) was then added, and incubated for 5 min. Stained cells were analyzed by flow cytometry. Cell death is expressed as % PI^+^.

### 2.12. Statistical Analysis

Statistical analysis was performed using ANOVA and paired Student’s *t*-test. A *p* < 0.05 was considered as statistically significant.

## 3. Results

### 3.1. 5-FU-Induced Upregulation of Fas Expression in Human Colon Tumor Cells Depends on p53

5-FU can activate p53 and upregulate Fas expression in human colon tumor cells [31,32,33,34]. To determine whether 5-FU upregulates Fas expression through activating p53, we made use of human colon tumor cells lines with WT and mutated p53. The rationale is that, if p53 is essential for Fas expression induction by p53, then tumor cell lines with mutated p53 should lose response to 5-FU, to upregulate Fas. Among the nine tumor cell lines analyzed, two lines (RKO and LS174T) were p53 WT, and the other seven lines harbored a p53 frameshift mutation and point of mutations (Appendix A). The human colon tumor cell lines were treated with 5-FU (5 μg/mL) and analyzed for Fas expression by flow cytometry. Only the RKO and LS174T cell lines exhibited increased Fas expression after 5-FU treatment (Appendix A). All of the other seven human colon tumor cell lines with mutated p53 [35,36,37] exhibited no response to 5-FU, in terms of Fas expression (Appendix A). Western blot analysis revealed that 5-FU treatment increased p53 protein levels in RKO and LS174T cells, in a dose-dependent manner (Appendix A). To strengthen this finding, we made use of a pair of WT and p53 KO human colon tumor cell lines of HCT116. 5-FU treatment upregulated Fas expression in HCT116-WT cells, but not in the HCT116-p53.KO cells (Appendix A). Consistent with the Fas expression increase, Western blotting analysis revealed that 5-FU treatment elevated the p53 protein level in HCT116-WT cells, but not in HCT116-p53.KO cells (Appendix A). These findings indicate that 5-FU activates p53 to upregulate Fas expression in human colon tumor cells in vitro. 

### 3.2. 5-FU Upregulates Fas Expression through p53 in MDSCs

It is known that MDSC homeostasis is regulated by Fas [22]. It is also known that MDSCs tend to downregulate Fas, to decrease sensitivity to apoptosis, to accumulate in tumor-bearing mice [23]. In addition, it has been shown that 5-FU selectively kills MDSCs in tumor-bearing mice [11]. We therefore hypothesized that 5-FU also activates p53, to upregulate Fas expression in MDSCs. To test this hypothesis, we used the MDSC-like J774M cell line model. The J774M cell line is a CD11b^+^Gr1^+^ cell line derived from the J774A.1 leukemia cell line. J774M cells exhibit potent inhibitory activity against T cell activation [26], and therefore phenotypically and functionally mimic MDSCs. J774M cells have WT p53 [38]. 5-FU treatment increased the p53 protein level in J774M cells in a dose-dependent manner, in vitro (Figure 1A). Consistent with the elevated p53 level, 5-FU treatment also upregulated Fas expression in J774M cells, in vitro (Figure 1B,C). To determine whether, like in the colon tumor cell lines, 5-FU upregulates Fas expression through activating p53, *Trp53* was knocked down in J774M cells, using the CRISPR technique (Figure 1D). Knocking down p53 significantly decreased the Fas protein level, in response to 5-FU treatment in J774M cells (Figure 1E,F). Our finding indicates that 5-FU activates p53 to upregulate Fas expression in MDSCs, in vitro.

### 3.3. 5-FU Increases MDSC Sensitivity to FasL Induced Cell Death

The above findings indicate that 5-FU treatment increases Fas expression in MDSCs. We next sought to determine whether increased Fas expression leads to enhanced MDSC sensitivity to FasL-induced cell death. J774M cells were treated with FasL either alone or in combination with 5-FU. Analysis of cell death revealed that 5-FU significantly increased J774M cell sensitivity to FasL-induced cell death, in vitro (Figure 2A,B). 

### 3.4. 5-FU Suppressses MDSCs Accumulation and Increases CTLs Infiltration In Vivo

To elucidate whether the finding observed in MDSC-like cells, in vitro, can be extended to MDSC suppression in tumor-bearing mice, in vivo, we made use of the CT26 colon tumor mouse model. CT26 cells have WT p53 [39]. As expected, 5-FU treatment increased the p53 protein level and Fas expression in a dose-dependent manner in CT26 tumor cells, in vitro (Appendix A). CT26 cells were then injected subcutaneously into BALB/c mice. The CT26 tumor-bearing mice were treated by 5-FU. As expected, 5-FU therapy significantly inhibited growth of the established CT26 tumor (Appendix A). Analysis of the tumor cells indicates that 5-FU therapy significantly increased Fas expression on tumor cells, and decreased CD45^−^ tumor cells accumulation in vivo (Appendix A). 

We then analyzed Fas expression and MDSC accumulation level in the tumor, spleen, and bone marrow of the CT26 tumor-bearing mice. Tumor tissues, spleens, and bone marrows were processed into single cells. CD11b^+^Gr1^+^ MDSCs were gated and analyzed for Fas protein level on the MDSC cell surface. 5-FU therapy significantly increased the Fas protein level on the tumor-infiltrating MDSCs (Figure 3A), spleen (Figure 3D), and bone marrow (Figure 3F). Consistent with the increased Fas protein level, 5-FU therapy suppressed splenomegaly (Figure 3C) and significantly decreased MDSC accumulation in the tumor (Figure 3B), spleen (Figure 3E), and bone marrow (Figure 3G) in the tumor-bearing mice.

MDSCs are potent suppressors of T cells. Our findings reported above, that 5-FU therapy upregulates Fas expression in MDSCs, to decrease MDSC accumulation in tumor-bearing mice, suggest that 5-FU therapy should also increase CTL activation, resulting in an increased level of tumor-infiltrating CTLs. To test this hypothesis, we analyzed tumor-infiltrating CTLs. It is clear that there are significantly more tumor-infiltrating CTLs in 5-FU-treated tumor-bearing mice than in the control mice (Figure 4A,B). 

### 3.5. Fas and FasL Expression Profiles in Human Colon Cancer

To determine the human cancer relevance of our above findings, we then analyzed the cellular source of Fas, in colon cancer patients. We mined scRNA-Seq datasets (GSE178341) [28] and analyzed Fas and FasL expression profiles in human colon tumors. Colon tumor-resident cells were annotated (Figure 5A). Cellular subtype analysis demonstrated that Fas is primarily expressed in T, NK, B, and stroma cells in human colon carcinoma (Figure 5A). Fas expression levels in myeloid cells and epithelial cells/tumor cells are weak, and it is only detectable in a small portion of myeloid cells (Figure 5B). 

As expected, FasL is highly expressed in T cells and NK cells (Figure 6A,C). Colon tumor-resident immune cells were also annotated (Figure 6B). Cellular subtype analysis demonstrated that FasL is highly expressed in CD8^+^CXCL13^+^ proliferating T cells, PLZF^+^ proliferating T cells, gd-like proliferating T cells, CD8^+^CXCL13^+^ HSP^+^ T cells, and NKCD16A^+^ NK cells (Figure 6B,C). 

### 3.6. 5-FU Chemotherapy Suppresses MDSCs Accumulation and Increases CTL Level in Human Colon Cancer Patients

Our above analysis indicates that FasL is highly expressed in T cells and NK cells, but Fas expression in myeloid cells is low. It is therefore expected that 5-FU chemotherapy should upregulate Fas expression in myeloid cells, to sensitize the myeloid cells to FasL-induced apoptosis by T cells and NK cells. We therefore compared MDSCs levels in the peripheral blood between human colon cancer patients without 5-FU chemotherapy and human colon cancer patients who received 5-FU chemotherapy. Flow cytometry analysis revealed that 5-FU chemotherapy did not significantly change the total CD11b^+^ cells (Figure 7A), but significantly decreased CD11b^+^HLADR^−^ MDSC-like MDSCs (Figure 7B). Furthermore, patients who received 5-FU chemotherapy had a significantly higher level of CTLs in their peripheral blood, as compared to untreated patients (Figure 7C). Our findings indicate that 5-FU chemotherapy suppresses MDSC accumulation and increases CTL level in human colon cancer patients.

## 4. Discussion

Myeloid cell homeostasis is controlled by the balance of myeloid progenitor cell differentiation into mature myeloid cells, and cell death [40]. It is well known that pro-inflammatory factors induce MDSC differentiation from myeloid progenitor cells in a tumor-bearing host [41]. Increased differentiation induced by tumor-secreted factors is therefore a major contributor of MDSC accumulation in human cancer patients and tumor-bearing mice [40]. However, it is also known that the massive accumulation of MDSCs in human cancer patients and tumor-bearing mice is due to the dysregulated cell death pathways [42,43,44,45]. Several cell death pathways have been identified in MDSCs, for their roles in MDSC turnover and accumulation [42,43,44,46,47,48,49,50,51]. The Fas–FasL cell death pathway was originally identified as a key regulator of T cell turnover [13], and dysregulation of the Fas–FasL pathway leads to massive accumulation of T cells [52,53,54,55]. However, the function of Fas has since been extended to myeloid cells, and it is known that Fas-mediated apoptosis regulates MDSC homeostasis [22]. Furthermore, MDSCs tend to downregulate Fas expression and deregulate the apoptosis-regulatory mediators Bax and Bcl-xL, which are downstream of the Fas receptor, to decrease MDSC sensitivity to FasL-induced apoptosis [22,23,24,25,26,27,28,29,30,31,32,33,34,35,36,37,38,39,40,41,42,43,44,45,46,47,48,49,50,51,52,53,54,55,56]. In this study, we observed that, as expected [13], FasL is highly expressed in T cells and NK cells in human colon carcinoma, but Fas expression is undetectable in most myeloid cells in the tumor microenvironment. Our findings thus suggest that, myeloid cells such as MDSCs may use attenuated Fas expression as a mechanism to avoid cell death induction by FasL of T cells in human colon cancer patients. Consistent with this notion, we observed that 5-FU increased Fas expression in MDSCs, and decreased MDSC accumulation in colon tumor-bearing mice. MDSCs are potent suppressors of T cells and NK cells [8,40], and it has been shown that 5-FU selectively kills MDSCs, to increase CTL level in tumor-bearing mice [11]. In this study, we also observed that decreased MDSC accumulation in 5-FU-treated tumor-bearing mice was correlated with increased T cell tumor infiltration. Furthermore, we demonstrated that 5-FU-based chemotherapy decreased the accumulation of MDSC-like cells, and increased T cell level in human colorectal cancer patients. Mechanistically, we determined that 5-FU activates p53 in MDSCs to upregulate Fas expression, to increase MDSC sensitivity to FasL-induced apoptosis. Our findings thus indicate that 5-FU chemotherapy exerts its efficacy, at least in part, through activating Fas expression in MDSCs, to suppress MDSCs, to increase T cell level in colon tumor-bearing mice and human colorectal cancer.

Chemotherapeutic agents such as 5-FU may cause DNA damage to activate p53 in human colon tumor cells [32,34]. p53 is a known Fas transcriptional activator [25,31,33,57]. In this study, we determined that 5-FU activation of Fas depends on p53. Among the nine human colon tumor cell lines, only the two tumor cell lines with the WT *TP53* gene responded to 5-FU, to upregulate Fas, indicating that p53 is essential for 5-FU’s function in activating Fas expression in human colon tumor cells. However, the *TP53* gene is mutated at the somatic level in more than 60% of human colorectal cancer patients [36,58]. The 5-FU–p53–Fas–FasL pathway can therefore only be activated in tumor cells with a WT *TP53* gene in human colorectal cancer patients. In addition to the death receptor Fas, the Fas-mediated apoptosis pathway also depends on the downstream signaling pathway [59,60], which is regulated by pro-apoptotic and anti-apoptotic proteins [2], such as the B cell lymphoma 2 (Bcl-2) family protein, inhibitor of apoptosis proteins (IAPs), and myeloid leukemia cell differentiation protein (Mcl-1) [61,62,63,64,65]. Colorectal tumor cells often deregulate expression of these apoptosis regulators, to acquire a resistant phenotype to 5-FU [66,67,68,69]. The p53–Fas–FasL tumor intrinsic pathway, thus, may only play a role for 5-FU chemotherapy efficacy in killing tumor cells through the Fas-mediated apoptosis, in a small portion of colorectal cancer patients. In contrast, MDSCs are transiently induced from bone marrow cells by tumor-secreted factors, and germline *TP53* mutation is rare in human colorectal cancer patients [70]. Although the intrinsic anti-apoptotic apoptosis regulator may be deregulated in MDSCs [23], the death receptor appears to be a dominant factor in apoptosis induction in hematopoietic cells, including MDSCs [23,43,50,71,72,73,74,75]. The p53–Fas–FasL pathways, therefore, in MDSCs, may be an important contributor of 5-FU efficacy in colorectal cancer.

Although CTLs infiltrates are present in human colorectal carcinoma [76], human colorectal cancer essentially does not respond to ICI immunotherapy [77]. MDSCs are potent suppressors of CTLs, and MDSCs suppress CTL activation through both PD-L1-dependent and independent mechanisms [78,79]. MDSCs may thus re-suppress PD-1 blockade-activated CTLs, leading to colorectal cancer non-response to PD-1 blockade immunotherapy. In this study, we demonstrated that 5-FU chemotherapy activates the intrinsic p53–Fas pathway in MDSCs to improve MDSC sensitivity to FasL-induced apoptosis in vitro, and suppress MDSC accumulation in colon tumor-bearing mice in vivo. We further determined that 5-FU chemotherapy suppresses MDSC accumulation, and increases the CTL level in human colorectal cancer patients. Our findings indicate that 5-FU chemotherapy might be an effective approach to enhance human colorectal cancer response to ICI immunotherapy.

One limitation of this study is that, the phenotypes of MDSCs in human colorectal cancer patients is not characterized. We have observed that the CD11b^+^HLADR^−^ MDSC-like cells are suppressed by 5-FU-based chemotherapy. More studies are needed to further characterize the 5-FU-suppressed MDSC phenotypes, and their role in CTL activation and tumor inhibition during 5-FU-based chemotherapy in human colorectal cancer patients. Another limitation of this study is that, the direct link between 5-FU-mediated suppression of MDSC accumulation and increased CTL level in tumor-bearing mice and human colorectal cancer patients is not established. Although MDSCs are well known to be potent CTL suppressors [8,40], it is possible that 5-FU may induce immunogenic cell death to increase CTL tumor infiltration [80,81,82], which requires further study. Nevertheless, our findings have extended the mechanism of 5-FU’s antitumor activity from direct tumor cell cytotoxicity to MDSC suppression, at least in part through the intrinsic p53–Fas pathway in MDSCs, to increase CTL level, to suppress colon cancer.

## 5. Conclusions

5-FU is the standard chemotherapeutic agent for human colorectal cancer treatment. The mechanism of action of 5-FU has been believed to directly induce tumor cell death through inhibition of DNA and RNA synthesis in tumor cells. Recent advance in myeloid regulatory cells revealed that 5-FU also selectively kill MDSCs in tumor-bearing mice. We determined that 5-FU activates p53 to upregulate Fas expression on MDSCs. 5-FU-induced Fas expression upregulation enhanced MDSC sensitivity to FasL-induced MDSC death. In addition, we observed that 5-FU therapy upregulated the expression of Fas on MDSCs in colon tumor-bearing mice to increase CTL tumor infiltration. Furthermore, we extended our finding in tumor-bearing mice to human cancer and determined that 5-FU chemotherapy suppressed MDSC accumulation and increased T cell level in human colorectal cancer patients. Our findings thus indicate that 5-FU activates the intrinsic p53-Fas pathway to suppress MDSC accumulation, which at least in part contributes to elevated CTL level and tumor inhibition. Our data also suggests that 5-FU chemotherapy is potentially an effective approach to overcome human colorectal cancer nonresponse to PD-1 blockade immunotherapy by suppressing MDSCs to reverse PD-L1-independent immune suppression mechanism.

## Figures and Tables

**Figure 1 cancers-15-01563-f001:**
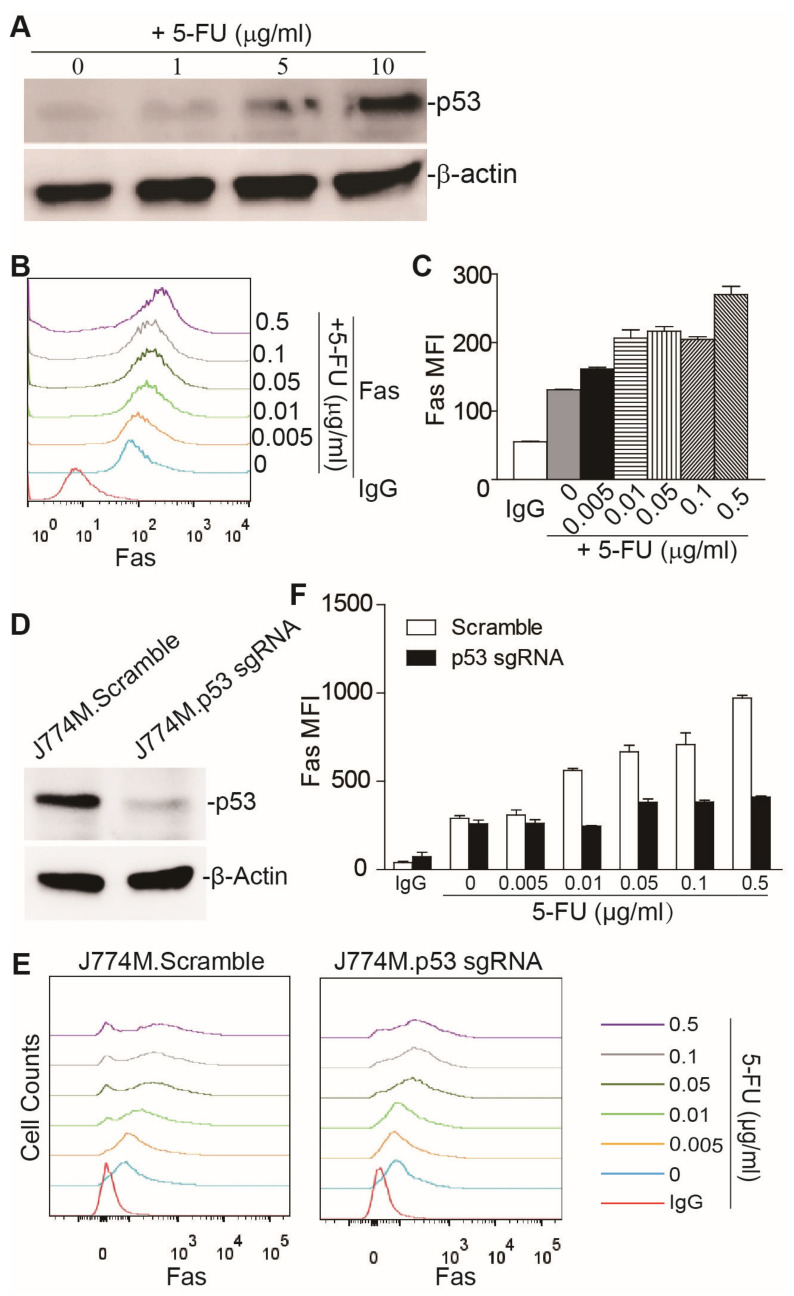
5-FU induces Fas expression in MDSCs, in vitro. (**A**) J774M cells were cultured in the presence of 5-FU at the indicated concentrations, for 24 h, and analyzed for p53 expression by Western blotting. (**B**) J774M cells were treated with 5-FU at the indicated concentrations, for 48 h, and analyzed for Fas expression by flow cytometry. (**C**) Quantification of Fas mean fluorescence intensity (MFI), from (**B**). (**D**) J774M.Scramble and J774M.p53.sgRNA cells were cultured in the presence of 5-FU for 24 h, and analyzed for p53 expression by Western blotting. (**E**) J774M.Scramble and J774M.p53.sgRNA cells were cultured in the presence of 5-FU at the indicated concentrations, for 48 h, and analyzed for Fas expression by flow cytometry. (**F**) Quantification of Fas MFI from (**E**).

**Figure 2 cancers-15-01563-f002:**
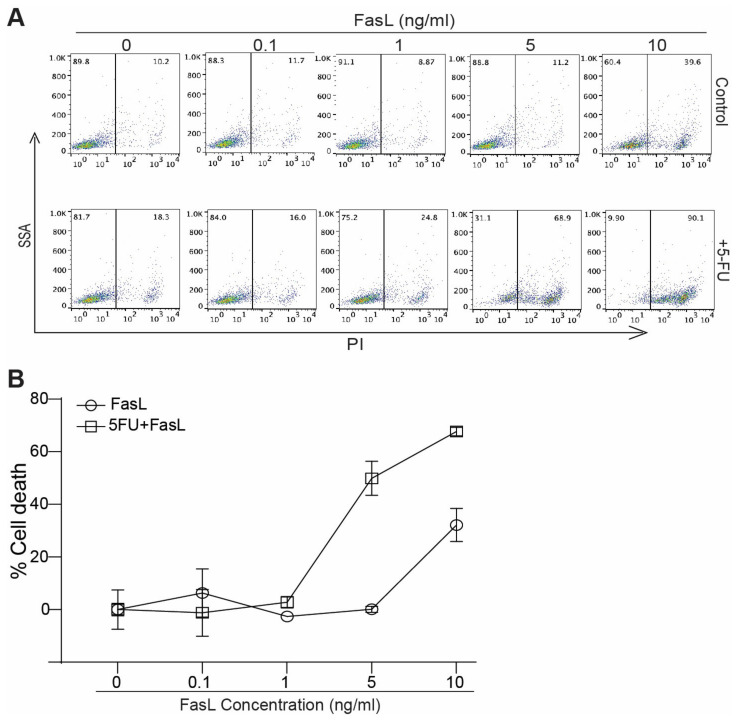
5-FU sensitizes MDSCs to FasL-induced apoptosis, in vitro. (**A**) J774M cells were treated by FasL or 5-FU (10 μg/mL) + FasL at the indicated concentrations, for 24 h. Cells were then analyzed by PI staining and flow cytometry. Shown are representative dot plots. (**B**) Quantification of percentage of cell death (% PI^+^ cells).

**Figure 3 cancers-15-01563-f003:**
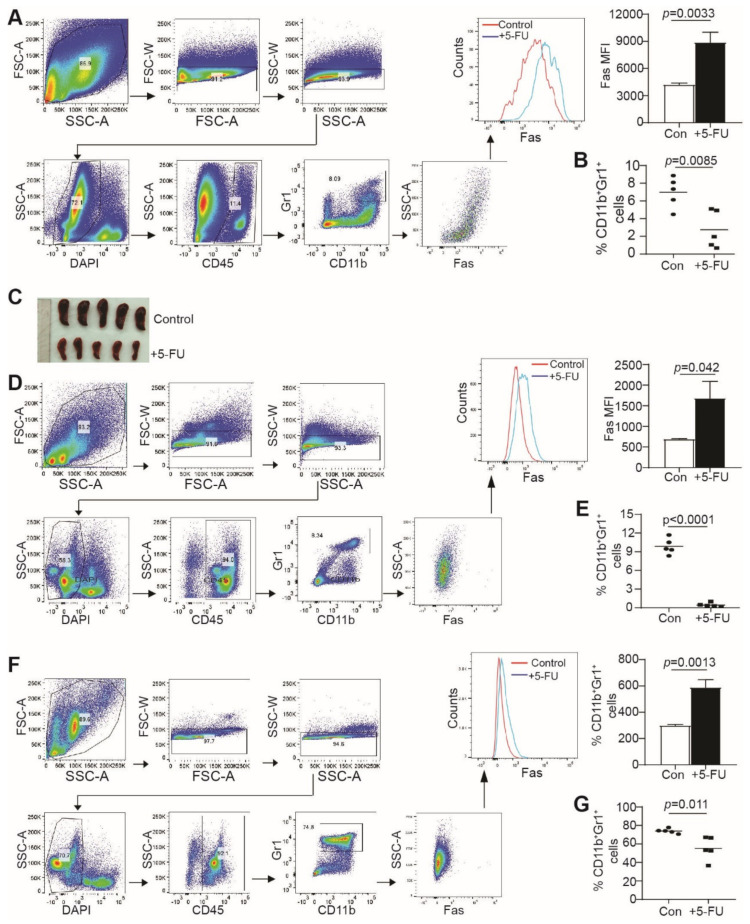
5-FU suppresses MDSCs accumulation, in vivo. (**A**) CT26 tumor-bearing mice were treated, as Appendix A. CT26 tumor tissues were digested into single cells and analyzed for Fas expression in tumor-infiltrated MDSCs, by flow cytometry. Shown on the left is the gating strategy, in the middle is the representative histogram of Fas, on the right is the MFI of Fas. (**B**) Quantification of CD11b^+^Gr1^+^ cells in CT26 tumor tissues. (**C**) Spleen image of tumor bearing mice for control and 5-FU-treated group. (**D**) Spleens of tumor-bearing mice were processed into single cells and analyzed for Fas expression in MDSCs by flow cytometry. Shown on the left is the gating strategy, in the middle is the representative histogram of Fas, on the right is the MFI of Fas. (**E**) Quantification of CD11b^+^Gr1^+^ cells in spleens. (**F**) Bone marrows of tumor-bearing mice were processed into single cells and analyzed for Fas expression in MDSCs, by flow cytometry. Shown on the left is the gating strategy, in the middle is the representative histogram of Fas, on the right is the MFI of Fas. (**G**) Quantification of CD11b^+^Gr1^+^ cells in bone marrows.

**Figure 4 cancers-15-01563-f004:**
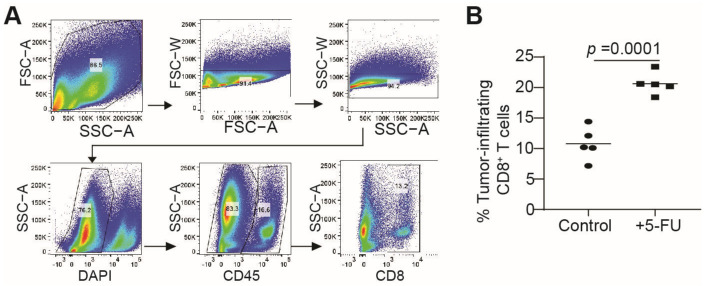
5-FU increases tumor-infiltrating CTLs, in vivo. (**A**) CT26 tumor-bearing mice were treated, as Figure 3. CT26 tumor tissues were digested into single cells and analyzed for CD8^+^ tumor-infiltrating T cells, by flow cytometry. Shown is the gating strategy. (**B**) Quantification of CD8^+^ tumor-infiltrating cells.

**Figure 5 cancers-15-01563-f005:**
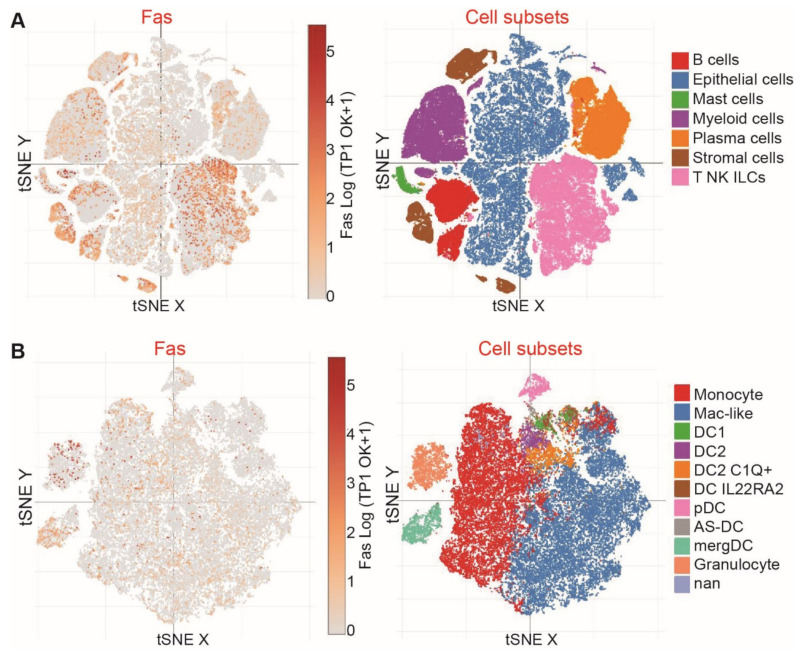
FAS expression profiles at the single cell level, in colon cancer patients. (**A**) scRNA-Seq datasets of human colon cancer patients were extracted from the Broad Institute Single Cell Portal. Shown are the tSNE plots of the subpopulations of cells, as indicated (right panel). FAS expression level in the specific cell population is shown in the left panel, tSNE. (**B**) scRNA-Seq datasets of human colon cancer patients were extracted from the Broad Institute Single Cell Portal. Shown is the tSNE of immune cell subpopulations (right panel). FAS expression level in the specific immune cell population is shown in the left panel.

**Figure 6 cancers-15-01563-f006:**
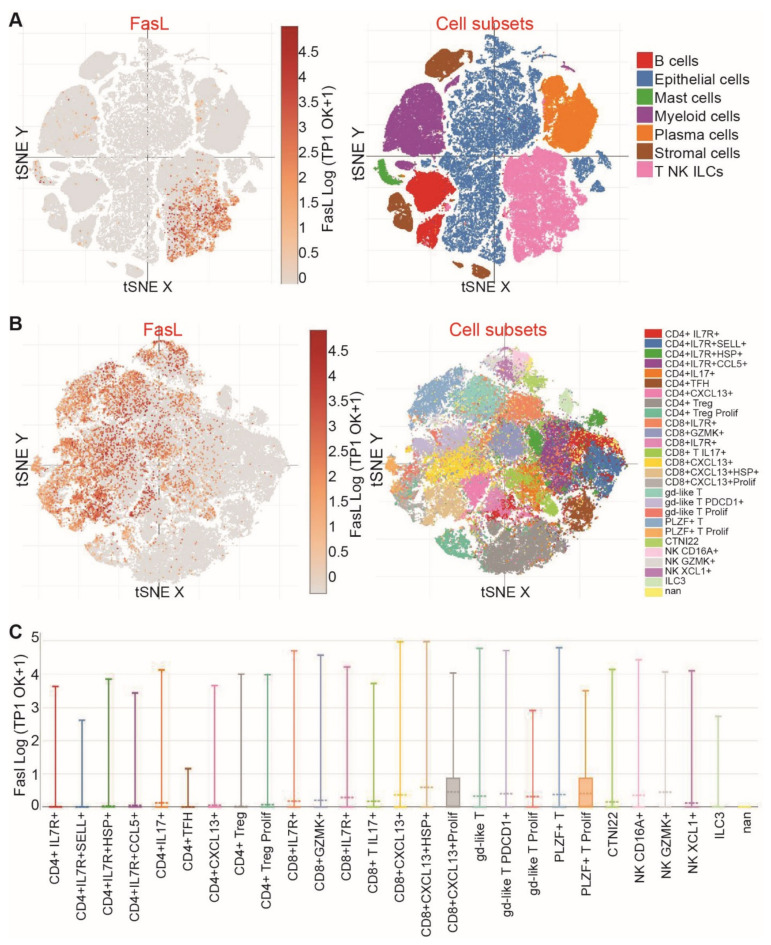
FASL expression profiles at the single cell level, in colon cancer patients. (**A**) scRNA-Seq datasets of human colon cancer patients were extracted from the Broad Institute Single Cell Portal. Shown are the tSNE plots of the subpopulations of cells, as indicated (right panel). FASL expression level in the specific cell population is shown in the left panel tSNE. (**B**) scRNA-Seq datasets of human colon cancer patients were extracted from the Broad Institute Single Cell Portal. Shown is the tSNE of immune cell subpopulations (right panel). FASL expression level in the specific immune cell population is shown in the left panel tSNE. (**C**) Expression of FasL in indicated immune cells.

**Figure 7 cancers-15-01563-f007:**
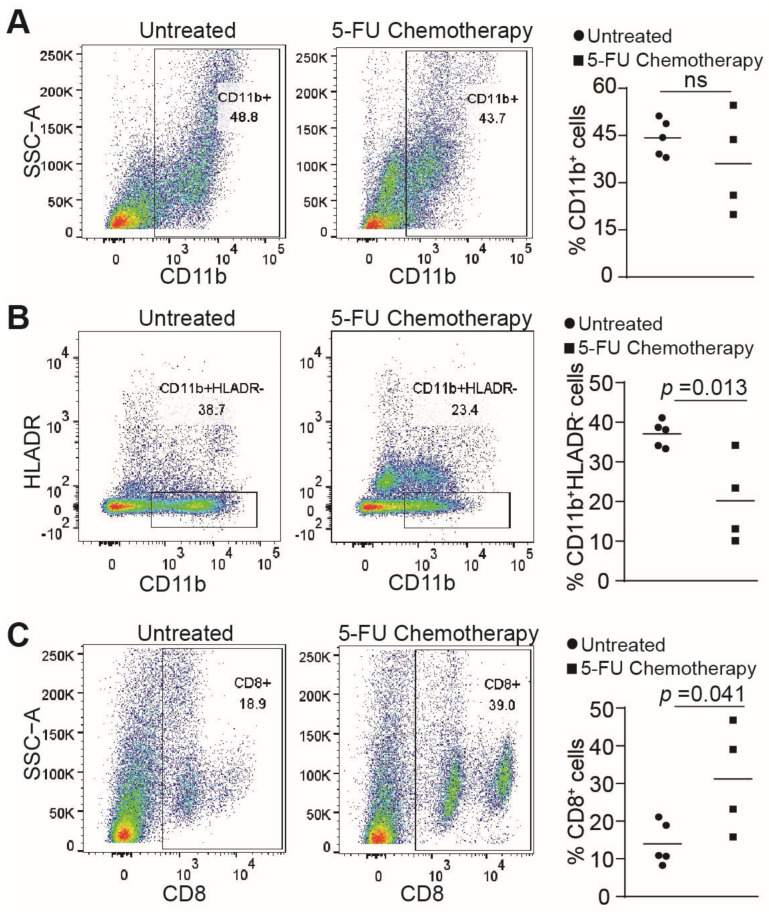
5-FU suppresses MDSCs accumulation and increases CTLs level, in human colon cancer patients. (**A**) Five untreated and four 5-FU-treated human blood samples from colorectal patients were processed into single cells and analyzed for CD45^+^CD11b^+^ cells, by flow cytometry. Shown on the left is a representative dot plot, and quantification of percentage is shown on the right. ns: no significance. (**B**) Human blood samples were processed as A and analyzed for CD45^+^CD11b^+^HLADR^−^ cells by flow cytometry. Shown on the left is a representative dot plot, and quantification of percentage is shown on the right. (**C**) Human blood samples were processed as A, and analyzed for CD45^+^CD8^+^ cells, by flow cytometry. Shown on the left is a representative dot plot, and quantification of percentage is shown on the right.

## Data Availability

Publicly available datasets were analyzed in this study. The datasets are described in the method and results section.

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
