# Peer review of "5-Fluorouracil Suppresses Colon Tumor through Activating the p53-Fas Pathway to Sensitize Myeloid-Derived Suppressor Cells to FasL+ Cytotoxic T Lymphocyte Cytotoxicity"

_cancers, 2023, doi:10.3390/cancers15051563_

Round 1

Reviewer 1 Report (Previous Reviewer 2)

The authors respond to my comments properly. There is no other concern.

Author Response

Point 1: The authors respond to my comments properly. There is no other concern.

Response 1: We would like to thank the reviewer for reviewing our manuscript again.

Reviewer 2 Report (Previous Reviewer 1)

Thank you for addressing all of the comments and for improving your manuscript. 

Please make a final language and spell check.

I can see now and understand the figures though I can still see the background but the exp graph and plot has fine resolution.

Author Response

Point 1: Thank you for addressing all of the comments and for improving your manuscript. 

Response 1: We would like to thank the reviewer for the suggestions and for reviewing our manuscript again.

Point 2: Please make a final language and spell check.

Response 2: We thank the reviewer for this advice. The authors and Ms Dakota Poschel checked the manuscript thoroughly and corrected all errors. We submitted the revised manuscript. The revised sections are in yellow fonts. Please see the attachments.

Point 3: I can see now and understand the figures though I can still see the background but the exp graph and plot has fine resolution.

Response 3: Yes, we improved the figure quality as suggested. We thank the reviewer for this advice. 

This manuscript is a resubmission of an earlier submission. The following is a list of the peer review reports and author responses from that submission.

Round 1

Reviewer 1 Report

1.     The manuscript needs a major language revision by a professional English speaker.

2.     There many repetitions in the same phrase and in adjacent phrases. Please correct and avoid. There are also many grammar errors.

3.     Please provide better link between end of paragraph and beginning of new paragraph, the story seems cut at times and not fluid and this makes it difficult to follow the story and understand

4.     Images are not understandable, not even easy to see, you need to provide high resolution images.

5.     Please provide a more professional and specific explanations of experiments in the Materials and Methods section for researchers need to be able to reproduce exactly your experiments by reading your Materials and Methods sections.

i.e. line 83 “…cancer patients with and without 5-FU…” maybe better as : …cancer patients untreated and treated with 5-FU…”

Do you specify concentration in the text?

i.e. line 114 “The tumor bearing mice…”:  Did you wait until the tumor reached what size to begin the treatment?

i.e line 136 “…For cell lines, cells were resuspended in 100 ul PBS….” Did you first harvest and centrifuge the cells to resuspend the pellet  in PBS?

i.e. line 137: there is typo, please correct all typos.

i.e. line 151 “taken” don’t think you can use this maybe “was considered”

6.     Line 156: can you better explain what mutations of p53 you are talking about and why they shouldn’t work: are they inactivating mutations or p53 activating mutations-be careful.

7.     Do you need to insert the rationale? Just explain your goal (demonstrate p53 dependence) and how you performed the experiments, what you observed as results. Did you think of using HCT116 p53wt vs syngeneic HCT116 P53-/-?

8.     “Western blotting analysis….” correct: “ Analysis of protein levels through western blot revealed…”

9.     Why p53 levels are not observed in RKO and LS174T basally?

10.  The “knock out” of p53 is actually a strong decrease since we can still see it, I would keep more mild since you do not observe a total removal of p53. Or maybe I don’t see it for bad quality of image.

11.  Lin199: were all tumors same size at beginning of treatment? What was the vehicle of 5-FU? Did you check the toxicity of vehicle?

12.  Lin203-209: did you check p53 levels?

13.  Check consistency of references.

Reviewer 2 Report

The authors tried to elucidate the underlying mechanisms why 5-fluorouracil (5-FU) can decrease myeloid-derived suppressor cells (MDSCs) using human colon cancer cell lines and a mouse colon cancer model. They utilized a J774M macrophage cell line as MDSCs in vitro. They show that 5-FU treatment increased the expression of p53 protein, as well as Fas expression, in J774M cells and knockdown of p53 abolished this effect on Fas. In addition, the combination of 5-FU and FasL induced higher levels of cell death in J774M cells compared when treated with FasL alone. In CT26-bearing mice, 5-FU treatment increased the Fas expression on cancer cells and decreased MDSCs in vivo, and increased tumor-infiltrating CTLs. scRNA-Seq datasets of human colon cancer patients show that Fas was expressed on T, NK, B and stromal cells, and FasL was expressed on T and NK cells. Finally, they show a decreased in MDSC and an increase in CTL in tumor tissues of colon cancer patients who received 5-FU. Although the conclusion may be plausible, direct evidence is not presented.

Specific comments:

1) The authors used J774M as a MDSCs in vitro. MDSCs purified from CT26-bearing mice should be used.

2) Figure 1E. The doses of 5-FU may be reversed.  

3) Throughout the paper, quality of the flow cytometry data is poor.  

4) How about the sensitivity of MDSCs treated with or without 5-FU treatment to FasL-expressing T cells or NK cells.

5) Figure 3. Immune cells other than MDSCs can be increased their Fas expression via p53 activation, too. Is there any specificity that 5-FU treatment induced Fas-mediated cell death only in MDSCs?

6) Figure 4. Cell death induced by any anti-cancer drugs could promote CTL infiltration as a result of cancer cell death. This observation is not always only due to a decrease of MDSCs.  

7) Figure 7. Decease of MDSCs after 5-FU treatment and increased infiltration of CTLs do not necessarily related to each other. This observation does not support that conclusion that 5-FU increased Fas expression on MDSCs and killed by FasL-expressing immune cells.